# Isolation, Maintenance and Differentiation of Primary Tracheal Basal Cells from Adult Rhesus Macaque

**DOI:** 10.3390/mps2040079

**Published:** 2019-10-01

**Authors:** Anna E. Engler, Gustavo Mostoslavsky, Lisa Miller, Jason R. Rock

**Affiliations:** 1Center for Regenerative Medicine of Boston University and Boston Medical Center, The Pulmonary Center and Department of Medicine, Boston University School of Medicine, Boston, MA 02118, USAgmostosl@bu.edu (G.M.); 2California National Primate Research Center, University of California, Davis, School of Veterinary Medicine, Davis, CA 95616, USA; lmiller@ucdavis.edu

**Keywords:** rhesus macaque, tracheal basal cells, primary culture, air–liquid interface, tracheospheres

## Abstract

In this report, we describe methodologies for the isolation and culture of primary rhesus macaque tracheal basal cells, their cryopreservation, long term storage and differentiation. These are comparable to state-of-the-art protocols that have been developed for mouse and human airway basal cells. This method is based on the use of proprietary media, providing an easily reproducible and applicable protocol for usage in biosafety level 2 (BSL2) settings. Tracheas from rhesus macaques were isolated after animal euthanasia and subjected to enzymatic digestion overnight. Cells of the epithelial layer were scraped off of the trachea for cell culture. Twenty-four hours after plating basal cells had attached and nonadherent cells were removed. First passages of basal cells can be frozen for early passage storage in liquid nitrogen or propagated and differentiated on an air–liquid interface and in a tracheosphere assay up to passage seven. This protocol provides a platform for the analysis of basal cells from a close evolutionary relative to humans.

## 1. Introduction

The pseudostratified tracheobronchial epithelium provides a mechanical barrier against noxious and infectious agents [1]. Approximately one third of the cells in the pseudostratified epithelium are basal cells (BCs). Numerous studies have demonstrated that BCs self-renew and give rise to luminal lineages (secretory and multiciliated cells), fulfilling the definition of adult tissue stem cells [2,3,4,5]. BCs from mice or humans can be isolated, maintained and differentiated in vitro [2,6]. Currently there are no protocols available for the isolation, maintenance or differentiation of primary basal cells from rhesus macaques. We have developed these protocols to facilitate the isolation and expansion of macaque BCs for molecular and functional characterization. 

## 2. Experimental Design

This protocol can be divided in 5 experimental stages: (1) Coating (Section 3.1) and media preparation (Section 5); (2) tissue dissociation and cell isolation (Section 3.2); (3) culture and maintenance (Section 3.3); (4) manipulation and differentiation of macaque BCs (mmuBCs) (Section 3.4); and (5) freezing, long-term storage and thawing (Section 3.5). This protocol has been optimized for the use of Pneumacult-Ex for maintenance and Pneumacult-ALI for differentiation, which are both proprietary media available from Stem Cell Technologies. 

### Materials

For a complete list, see Appendix A. PureCol (Advanced Biomatrix, 5005-100ML, San Diego, USA)DNAse (Sigma DN25-100MG, Merck KGaA, Darmstadt, Germany)Protease (Sigma P5147-100MG, Merck KGaA, Darmstadt, Germany)DMEM/F12 (Gibco, 11330-032, Dublin, Ireland)Pneumacult-Ex (Stemcell Technologies, 05008, Cambridge, USA)Pneumacult-ALI (Stemcell Technologies, 05001, Cambridge, USA)Phosphate-buffered saline (PBS) (Gibco, 14190-144, Dublin, Ireland)Phosphate-buffered saline (PBS) (containing CaCl and MgCl) (Gibco, 14040-133, Dublin, Ireland)3D Matrigel (Matrigel Matrix Basement Membrane, Corning, 356231, New York, USA)

## 3. Procedure

Airway segments were collected at the time of necropsy from rhesus macaques with healthy lungs. Tissue was from adult rhesus macaque monkeys, age 2.5 years, 3.5 years and 8.5 years old. Cells obtained from animals older than 9 years did not yield sustainable cultures (Appendix A). Macaque basal cells are cultured on collagen coated plates (Section 3.1). The coating functions as a method of selection to purify the basal cells from the cell isolate after tissue dissociation (Section 3.2). After attachment and selection, the cells can be maintained and amplified in proprietary media without coating (Section 3.3). Under maintenance conditions, the cells do not differentiate, and we thus have optimized differentiation conditions for the macaque derived basal cells, commonly used in the field (Section 3.4), namely air–liquid interface differentiation (Section 3.4.1) and differentiation in tracheospheres (Section 3.4.2). mmuBCs can further be analyzed and manipulated by common techniques such as viral transfection (Section 3.4.3) and flow cytometric analysis and sorting (Section 3.4.4). For long term storage, cells can be cryopreserved (Section 3.5). For more details on expected results, yields and outcomes per section, we would like to refer to Section 4, Table 1, and for troubleshooting Section 4, Table 2. 

### 3.1. Coating: Time for Completion 02:30 h

Prepare 5 mL of coating medium (see Section 5) per 10 cm plate and pipet onto plate and ensure even distribution.

Incubate plate for 1–2 h at 37 °C.

Remove coating medium from plate and let dry for 30 min in a laminar flow hood.

Once dried, seal plates with Parafilm.
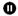
**PAUSE** Coated plates can be stored up to 2 weeks at 4 °C.

### 3.2. Tissue Dissociation and Cell Isolation: Time for Completion 20 h

Cut trachea into 3–4 smaller pieces using surgical scissors or a scalpel.

Dissect the upper, large airways out of the macaque lungs, ensure that as little lung tissue remains on these airways as possible.

Put pieces into clean wash buffer.

Remove any remaining lung, fat, connective and lymphoid tissue.

Cut the airway in 0.5 cm pieces and open the airway with a single longitudinal cut.

Transfer the pieces into a 50 mL conical tube containing 20 mL of cold digestion buffer.

Seal tube with paraffin, put tube into sealable plastic bag and incubate overnight for up to 16 h at 4 °C with gentle agitation. 

Inactivate the digestion buffer with 2mL of FBS (to a final concentration of ~10% FBS). The tissue will appear swollen but will still be intact. 

Remove pieces from conical tube and put into a 10 cm dish containing wash buffer and scrape off epithelial layer with a scalpel. Keep the digest medium on the side. 
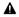
**CRITICAL STEP** Scraping should be done carefully, not cutting into the underlying stroma. We recommend holding a corner of the tissue with forceps and then stroking with the scalpel in a single direction, with a movement similar to spreading butter on toast. Discard remains. 

Combine wash buffer with scrapings with digest medium together and centrifuge for 5 min, at 500× *g*. 

Resuspend cells in 2–3 mL of Pneumacult-Ex (containing Penicillin, Streptomycin and Amphotericin B) and count cells or collect pellet for further analysis. 

Typically, 1 × 10^6^ cells can be harvested per 2–3 g of starting material.

Cells are counted using an automated counter (Luna II).

### 3.3. Culture and Maintenance: Time for Completion 4 to 5 days

For resuspension of the pellet, pipette up and down gently 3–4 times. 


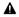
**CRITICAL STEP** Do not filter the resuspended pellet, BCs will attach, non-basal cells will not and can be washed off the next day. Cell yield is markedly increased if not filtered. 

Plate between 2–6 × 10^6^ cells/10 cm dish (3–11 × 10^4^ cells/cm^2^).

After plating, leave cells undisturbed for a minimum of 24 h.

The next day, remove medium, wash cells gently with PBS (containing CaCl and MgCl) and add fresh, warm medium.

Change medium every 48–72 h. Medium needs to be removed completely and cells are washed 1× with warm PBS (containing CaCl and MgCl) and 8–10 mL of fresh, warm Pneumacult-Ex are added. 

Reduce antifungal drugs gradually after 10–14 days in culture, by reducing the concentration 10-fold with every medium change. 

Due to clonal expansion, macaque BC cultures will never reach 100% confluency, subculture at 80% confluence. If cultures are grown for long time at maximum confluency, they will become quiescent. **OPTIONAL STEP** After the first passage cells do not need to be passaged onto coated dishes (see Section 3.1). For the first passage, after flow sorting, cell manipulations and after thawing collagen plates are an essential requirement.

Remove old medium but keep for later use.

Wash 2× with warm PBS.

Add 1mL of 0.25% trypsin per 10 cm plate. Scale down trypsin accordingly to plate size. 

Incubate for 3–7 min at 37 °C. Earlier passages will detach faster (3 min) than later passage numbers (up to 10 min maximum).

Gently tap dish and check under microscope that cells have detached, maximally incubate for 10 min in trypsin. Viability and yield tend to drop after 10 min in trypsin. 

Stop reaction adding 2 mL of FBS and wash plate with old medium. 

Centrifuge cells for 5 min at 500× *g.*

Remove supernatant and resuspend in 1 mL of fresh medium.

Count cells.

Plate 0.5 × 10^6^ cells per 10 cm plate in 10 mL fresh medium per plate. **OPTIONAL STEP** as few as 0.1 × 10^6^ cells can be plated in a 10 cm dish, initial growth will be slower. If even lower numbers of cells are plated, conditioned medium is highly recommended (see Section 5).

### 3.4. Manipulation and Differentiation: Time for Completion-Variable

For staining purposes, plate ~1 × 10^4^ cells per well of a chamber slide and fix the day after in 4% paraformaldehyde (PFA) for 20 min.

Wash gently 3× with PBS.

Keep in PBS at 4 °C until needed.

For staining of cells on chamber slide, it is recommended to stain directly within the chamber slide and only remove the grid after completion of staining. A more detailed staining protocol can be found in Appendix A. 

#### 3.4.1. Differentiation on Air–Liquid Interphase (ALI): Time for Completion 14 days 

Coat ALI insert for 30 min with coating medium in the incubator, remove the excess and afterwards dry for 30 min in a laminar flow hood. 

Dissociate and count cells (see Section 3.3).

Plate at 5 × 10^4^ cells/24-well insert density.

Add 400 µL of Pneumacult-Ex to the well and 200 µL to the cell culture insert. 

Culture cells according to maintenance protocol until confluency is reached and a cobblestone pattern is visualized. 

Once cells have reached confluency on insert, change medium from Pneumacult-Ex to Pneumacult-ALI medium in a gradual fashion. 


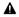
**CRITICAL STEP** Gradual change from Pneumacult-Ex to Pneumacult-ALI is highly recommended as cell layers tended to rupture or detach if an immediate change is performed. Gradual change results in more even, healthier cultures. For optimal culture health, media should be changed every 48 h at this point, until in 100% Pneumacult-ALI. 

Day 1 on ALI: Remove the medium from the insert and add fresh medium to the bottom of the well (400–500 µL). The insert is submerged in the medium allowing nutrients to diffuse through the lower membrane while the culture is exposed to the air.

On day 1 replace medium with 70% Pneumacult-Ex and 30% Pneumacult-ALI. 

On day 3 replace medium with 50% Pneumacult-Ex and 50% Pneumacult-ALI. 

On day 5 replace medium with 30% Pneumacult-Ex and 70% Pneumacult-ALI.

On day 7 replace medium with 100% Pneumacult-ALI medium. 

Maintain cells thereafter in Pneumacult-ALI medium. 

Completely replace Pneumacult-ALI medium every 48–72 hours. 

If ALI cultures are to be analyzed, fix and stain the cultures on the insert. For fixation, the medium in the bottom is removed and PBS is added. The cells are put back into PBS for 2–3 min. The PBS in the bottom is removed, this step is repeated 1–2 times. After washing, 4% PFA is added to the bottom of the well and to the insert for 5–10 min. 

Cut the insert from the transwell or stain whole insert within the transwell and cut after staining. A more detailed staining protocol can be found in Appendix A. 

#### 3.4.2. Differentiation in Tracheosphere Assay: Time for Completion 14 days

Dissociate and count cells (see Section 3.3).

Thaw 3D Matrigel on ice.

Resuspend cells in cold Pneumacult-Ex at a concentration of 100 cells/µL. 

Add cell culture inserts or small cell culture grade glass cover slips to plates (no coating needed) and UV irradiate for 5–10 min. 

Mix cells in 3D Matrigel 50:50 (to a final cell concentration of 50 cells/µL).

Add 90 µL of Matrigel/Cell Mix onto the insert or the glass coverslip. 
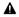
**CRITICAL STEP** Matrigel polymerizes quickly once at room temperature. We recommend using cold media, and doing all steps on ice. We usually use ice cold pipette tips for this step. Avoid air bubbles in the 3D Matrigel. 

For plating on insert, tap the plate after every 90 µL drop that is added to insert, so that there is an even layer. For plating on glass coverslip, let the drop polymerize without spreading.

Let Matrigel polymerize for 15–20 min in the incubator before adding medium. 

For the first 5 to 7 days the cells are cultured in Pneumacult-Ex. 

Feed cells every other day with Pneumacult-Ex. 

Once multicellular spheres are visible in the microscope (after 5–7 days in culture), transition the medium gradually into differentiation medium. 

For differentiation, feed cells for 2 days with 50% Pneumacult-Ex and 50% Pneumacult-ALI and afterwards with 30% Pneumacult-Ex and 70% Pneumacult-ALI. Replace medium completely every 48–72 h. 
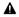
**CRITICAL STEP** The tracheospheres should not be moved to 100% Pneumacult-ALI. They are ultimately cultured in 30% Pneumacult-Ex and 70% Pneumacult-ALI medium. Sphere morphology is better in in this 30:70 medium compared to 100% Pneumacult-ALI. 

If the cells are not moved to Pneumacult-ALI medium, they will not differentiate. If maintained for too long in Pneumacult-Ex they will become solid spheres of Krt5+ basal cells. 

For fixation, resuspend the Matrigel in 4% PFA by pipetting up and down. 

After 10–15 min, transfer the slurry of PFA, Matrigel and Spheres into a 1.5 mL Eppendorf tube and fill with PBS (containing CaCl and MgCl). 

Centrifuge spheres (300× *g*, 3 min) and wash the pellet twice with PBS (containing CaCl and MgCl).

Stain spheres as whole mount if they are cultured for less than 14 days, afterwards the spheres become too dense and impermeable to the antibodies. After 14 days in culture, the tracheospheres will need to be sectioned to perform staining. A more detailed embedding and staining protocol can be found in Appendix A. **OPTIONAL STEP** For tracheosphere cultures at clonal density (5–10 cells/µL), the cells need to be cultured in a minimum of 30% conditioned Pneumacult-Ex Medium (see Section 5). They will take longer to form spheres (usually 10 days). The transition into differentiation medium should not be done before 14 days in vitro.

#### 3.4.3. Transduction of Macaque BC Cultures: Time for Completion 72 h

To demonstrate ability to transduce macaque basal cells, we utilized a preexisting lentivirus that had been developed for infecting human cells. 

Plate between 0.5–1.0 × 10^6^ cells/10 cm dish (0.75–1.5 × 10^4^ cells/cm^2^).

After plating, leave cells for a minimum of 24 h before further manipulation. 

For transduction, dilute virus of choice in Pneumacult-Ex and keep on ice. 

Remove medium from macaque BC cultures, wash once with PBS (containing CaCl and MgCl). Add medium containing high MOI of virus (MOI > 100) to adherent cells. **OPTIONAL STEP** Do not change medium before infection but use fresh medium to prepare virus to increase viral uptake.

Twenty-four hours after transduction, but no later than 36 h, wash the medium off the cells and replaced with fresh Pneumacult-Ex. 

Using a Lentiviral construct containing GFP under the CMV promoter, GFP expression is observed as early as 24 h post infection and will increase over the next 72 h (see Appendix A and Figure 4). 


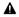
**CRITICAL STEP** Infected macaque BCs should be used within 7–14 days for experiments because transgene silencing reduced GFP expression. Even after sorting expression can be reduced by up to 60% within 14 days. In order to avoid silencing and establishing stable lines, Lenti-SIV virus constructs should be used [7].
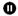
**PAUSE** GFP+ infected cells can be frozen (see Section 3.5) and thawed at a later time point. 

#### 3.4.4. Sample Preparation for Flow Cytometric Analysis or Purification

##### Flow Cytometric Analysis: Time for Completion 2 h

Dissociate cells (see Section 3.3) and resuspend in 1mL of 4% PFA in a 15 mL conical tube for 5 min. 

Wash out PFA by adding 13 mL of PBS (containing CaCl and MgCl) and centrifuge (300× *g*, 5 min). 

Resuspend cells in 1 mL of PBS (containing CaCl and MgCl) in a 15 mL conical tube. 

Gently agitate cells on a vortex while adding 100% ethanol dropwise to a volume of 10 mL (resulting in 90% ethanol). 

Keep cells at −20 °C for 30 min. 


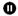
**PAUSE** Fixed and ethanol permeabilized cells can be stored in 90% ethanol for up to 5 days at −20 °C. 

For further staining, cells are centrifuged (500× *g*, 3 min) and washed twice with Flow Cytometry Buffer (see Section 5) followed by a 30 min blocking step in Flow Cytometry Buffer and stained with antibodies of interest thereafter. A more detailed protocol can be found in Appendix A. 

##### Purification by Flow Sorting: Time for Completion 30 min

Prepare fresh collagen plates (see Section 3.1).

Dissociate cells (see Section 3.3) and resuspend 1 × 10^6^ cells/100 µL of Flow Cytometry buffer. 

Add Calcein Blue at a concentration of 1 µL/mL of a 10 µM stock. 

Filter cells through the cell strainer cap of polystyrene round bottom tubes (40 µm mash). 

Sort cells for endogenous GFP expression (after infection) or cell surface markers, as well as live-dead stain (Calcein Blue). 

Sort cells into Conditioned Medium (see Section 5) containing 10 µM of Rho-kinase inhibitor Y-27632. 


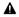
**CRITICAL STEP** Collection in conditioned medium containing Y-27632 and the usage of freshly prepared collagen-coated plates resulted in better recovery of the sorted cells. 

Plate sorted cells on freshly prepared collagen plates in conditioned medium containing Y-27632 or use for further downstream analysis (Single Cell Sequencing, RNA isolation, fixation). 

If the cells are plated after sorting, change the medium 24 h after plating to conditioned medium without Y-27632. After another 48 h the medium is changed to fresh Pneumacult-Ex medium. 

### 3.5. Freezing, Long Term Storage and Thawing

#### 3.5.1. Freezing and Long-Term Storage: Time for Completion 30 min 

Cells are dissociated and counted (see Section 3.3). 

Freeze a minimum of 5 × 10^5^ cells per vial.

Instead of plating, resuspend cells in 900 µL of conditioned medium (see Section 5) and lastly add 10% DMSO to cryovials. 

Invert cryovials 5–10 times gently.

Freeze cells slowly using isopropanol boxes at −80 °C and transferred to liquid nitrogen 48 h later. 

Cells can be kept at −140 °C for long-term storage and can successfully be thawed at least 1.5 years after freezing. 

#### 3.5.2. Thawing: Time for Completion 30 min

Thaw quickly in a water bath. 


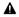
**CRITICAL STEP** To increase the cell viability and yield after thawing, PBS should be used cold and medium should be warmed no higher than room temperature. 

Resuspend in 10 mL of cold PBS (containing CaCl and MgCl). 

Centrifuge for 5 min, at 300× *g*, at 4 °C.

Remove PBS and resuspend in 10 mL fresh medium.

After thawing, plate cells on collagen coated plate. With the next passage they can again go onto uncoated dishes. 

## 4. Expected Results

This protocol should yield robust, primary cultures of macaque BCs within 2 days. These cells can be maintained for up to 7 passages and can be differentiated and manipulated as here described. 

### 4.1. Tissue Dissociation, Cell Isolation and Maintenance 

Work with rhesus macaque tissue and cells is performed at BSL2. Rhesus macaque monkeys are carriers for Hepatitis B and can be host species for various pathogens humans are susceptible to [8]. In order to obtain macaque BCs in culture, the trachea and/or upper airways were isolated (Figure 1a) and cut into smaller pieces (Figure 1b). The overnight digest was followed by mechanical scraping and cells were collected and plated onto collagen coated dishes. Freshly plated cells will be rounded and can be found in floating clumps (Figure 1c). The macaque BCs attach overnight, and the remaining dead cells and debris can be washed off, resulting in a monolayered culture of macaque BCs on collagen-coated plastic (Figure 1d). 

In order to assess the purity of cultures, we performed immunocytochemistry on early passage macaque BCs (Figure 2a). BCs from macaques, like BCs from mice and humans, express Krt5 (variably), Krt14 and TP63 [2,9]. The cells should be proliferative, marked by incorporation of EdU (Appendix A) (Figure 2b). Proliferative, BC cultures can be further used for differentiation assays (Section 4.2) and in culture transgenic manipulation (Section 4.3). 

### 4.2. Differentiation

When plated on cell culture inserts, cells will continue to proliferate and need to reach confluency. Transition from submersion to ALI conditions is done once a cobblestone morphology is visualized on the insert (Figure 3a, top). Once transitioned into ALI conditions the cells will start to differentiate and give rise to ciliated cells (Figure 3a, bottom). Cultures will rarely give rise to secretory cells under the conditions described. Alternatively, fresh macaque BC cultures can be differentiated in a tracheosphere assay [2]. Cells will form spheres in culture (Figure 3b, left panel) and once put into differentiation will start to give rise to Krt8^+^ progenitor cells and acet. Tubulin^+^ cells (Figure 3b, right panel). 

### 4.3. Macaque BC Culture Manipulations

Macaque BCs can be manipulated using Lentiviral transduction approaches (Figure 4). Macaque BCs can be transduced using Lentiviral constructs (Figure 4a) and thereafter sorted for transgene expression (Figure 4b). Infected macaque BCs exhibit waning GFP expression, even after sorting by up to 60% with high levels of transgene silencing (data not shown). In order to avoid silencing and establishing stable lines, Lenti-SIV virus constructs need to be used [7].

### 4.4. Anticipated Yields and Outcomes

On average the following yields were achieved from three distinct macaque BC (*n* = 3) cultures with experimental duplicates (*n_exp_* = 2), if not otherwise indicated. 

**Table 1 mps-02-00079-t001:** Expected yields.

	Expected Yield
Cell number from first dissociation	1.6 ± 0.2 × 10^6^ cells total (*n* = 3, *n_exp_* = 1)
Viability from first dissociation	>85% viability (*n* = 3, *n_exp_* = 1)
Cell number per 10 cm dish	3.5 ± 1.8 × 10^6^ cells total
Viability from later passages	>95% viability
Passage number till abnormal phenotypes occur	8.3 ± 3.5 passages
Time to form spheres	4.6 ± 0.6 days
Time to see cobblestone morphology on ALI	2.6 ± 1.2 days
Recovery and viability after thawing	67.7% ± 8.8% recovery, >65% viability
Cells in S-Phase (by EdU)	15.1% ± 4.5%
Average number of Krt5+ cells	99.2% ± 0.3%
Average number of Krt5+p63+ cells	89.7% ± 4.8%

### 4.5. Troubleshooting 

**Table 2 mps-02-00079-t002:** Troubleshooting.

Problem	Potential Cause	Solution
Low cell number from tissue	Short incubation in digest buffer Old/inactive digest buffer Poor scraping Tissue was from lower airways	16 h digest time Fresh digest buffer Gentle, but thorough scraping Use upper airway or trachea
Low cell number from first dissociation	Too short or too long trypsinization Trypsinization at RTC Too long in FBS solution after trypsinization	Monitor first dissociation carefully Trypsinize in incubator Immediately centrifuge and wash the cells
Low viability from first dissociation	Too long trypsinization Too much mechanical shearing	Monitor trypsinization carefully, stop when cells detach Gentle tapping/knocking the plate instead of pipetting
Low cell numbers after few passages and/or low viability	Too short or too long trypsinization Trypsinization at RTC Too long in FBS solution after trypsinization Too much mechanical shearing Cells become trypsin resistant	Monitor trypsinization carefully, stop when cells detach Trypsinize in incubator Immediately centrifuge and wash the cells Gentle tapping/knocking the plate instead of pipetting If cells do not detach after 10 min, they are abnormal and should be trashed
Poor attachment after dissociation	Collagen plate was prepared too long ago Collagen plate was not prepared correctly Plate was not coated	Collagen plates are good for maximum 2 weeks in the fridge Collagen plate needs to be prepared to contain 40 µg Collagen/mL Plate was not coated, dissociation was too harsh and cells need coating. Transfer non-attached cells to a coated plate
Low proliferation	Cells have not been dissociated frequently enough Primary cells are losing their proliferation capacity	Cells need to be dissociated frequently, when reaching 80% confluency to maintain a proliferative primary culture Primary cultures expire after ~9 passages.
Low number of Krt5+ cells	Digested for too long Scraping was too harsh Wrong side of the tissue scraped	Cultures need to be prepared from fresh tissue
Abnormal Morphology	Cells look spikey Cells look as though they have holes/vacuoles in them	Rho kinase inhibitor can make cells look spikey Early cells can look as though they have holes, they usually recover after few days Cells, after thawing look as though they have holes, they should recover
Poor recovery/viability after thawing	Cells were stored at -80 for too long Cells were frozen too fast Cells have been in liquid nitrogen for a long time	We recommend transfer to liquid nitrogen after 48 h We recommend, if more than 1 vial is frozen, to do a thawing control before storing cells for long term We have observed drop in recovery after cells were in liquid nitrogen for over 2 years, however viability should still be good.

## 5. Reagents Setup

These chemicals and products are commercially available. For a complete list of vendors and catalogue numbers see Appendix A. 

### 5.1. Coating Medium 

66 µL (200 µg) of bovine collagenase (PureCol, Advanced Biomatrix, 5005-100ML, 3 mg/mL) in 5 mL of PBS to a final concentration of 40 µg/mL per 10 cm plate

Coating medium needs to be prepared fresh for every use. Diluted and stored collagen is not recommended. The PureCol collagen stock is good in the fridge for up to 1 year. 

### 5.2. Wash Buffer 

500 mL of DMEM/F12 

5 mL of Penicillin Streptomycin 

1 mL Amphotericin C (Fungizone) (25 ng/mL)

Wash buffer is stable at 4 °C for 3–4 weeks. Ideally wash buffer is prepared fresh every time to ensure maximum protection from fungal or bacterial contamination in the cultures. 

### 5.3. Dissociation Buffers

#### 5.3.1. 10× Protease Mix

100 mg Protease (Protease from Streptomyces griseus) 

1 mg DNAse 

9 mL of cold PBS

The 10× Protease Mix should be frozen down as 2 mL aliquots immediately after preparation. Preparation should be done on ice with cold PBS. Protease Mix is stable at −20 °C for up to two years. 

#### 5.3.2. Digest Buffer

2 mL 10× Protease Mix (Section 5.3.1) 

0.2 mL Amphotericin B (2.5 µg/mL) 

17.8 mL cold Wash Buffer (Section 5.2)

Digest Buffer needs to be prepared fresh before each digest. Wash buffer should be cold when preparing the digest buffer. 

### 5.4. Pneumacult-Ex and Pneumacult-ALI

Pneumacult-Ex and Pneumacult-ALI are prepared fresh before every feed according to the manufacturer’s instructions. 

#### 5.4.1. Pneumacult-Ex

49 mL Pneumacult-Ex base medium

1 mL Pneumacult-Ex supplement 

100 µL Primocin

Pneumacult-Ex supplement is aliquoted upon arrival into 1 mL aliquots and stored at −20 °C. Pneumacult-Ex base medium is stored at 4 °C. The supplemented, complete medium is good for up to 2 weeks if stored at 4 °C and repeated warming and cooling is avoided. 

#### 5.4.2. Pneumacult-ALI

44.5 mL Pneumacult-ALI base medium

5 mL 10× Pneumacult-ALI supplement 

0.5 mL 100× Pneumacult-ALI maintenance supplement

100 µL Primocin 

Pneumacult-ALI 10x Supplement is aliquoted into 5 mL aliquots upon arrival. 100× Supplement is stored at −20 °C and protected from light. Pneumacult-ALI base medium is stored at 4 °C. The supplemented, complete medium is good for up to 2 weeks if stored at 4 °C, in the dark and repeated warming and cooling is avoided. 

### 5.5. Conditioned Medium

To prepare conditioned medium macaque BCs need to be cultured with fresh Pneumacult-Ex for 24–48 h. The medium is collected off the cultures and centrifuged at 1000× *g* for 5 min. The medium is collected, except the bottom 2 mL in the tube. This medium is then filtered through a 0.2 µm filter and stored at 4 °C for maximally 4 weeks. Before conditioned medium is used on the cells, the conditioned base medium is supplemented with Pneumacult-Ex supplement at a ratio of 1:20 (instead of 1:10). 

### 5.6. Flow Cytometry Buffer

49 mL PBS (containing CaCl and MgCl) 

0.5 mL Penicillin Streptomycin 

0.5 mL Amphotericin B (2.5 µg/mL)

0.5 g Bovine Serum Albumin 

Flow Cytometry Buffer is stable for 2 weeks at 4 °C. 

## Figures and Tables

**Figure 1 mps-02-00079-f001:**
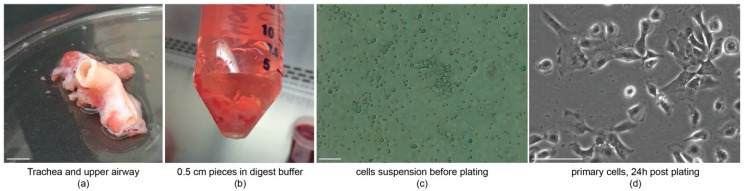
Cell isolation and plating. (**a**) Dissected trachea and upper airway for culture preparation, scale bar = 1 cm; (**b**) trachea and upper airway are opened longitudinally and cut into 0.5 cm pieces and digested over night; (**c**) cells are put into suspension before plating scale bar = 50 µm; and (**d**) basal cells will attach within 24 h after plating, and debris and non-basal cells can be washed off, scale bar = 20 µm. *n* = 3, *n_exp_* = 1.

**Figure 2 mps-02-00079-f002:**
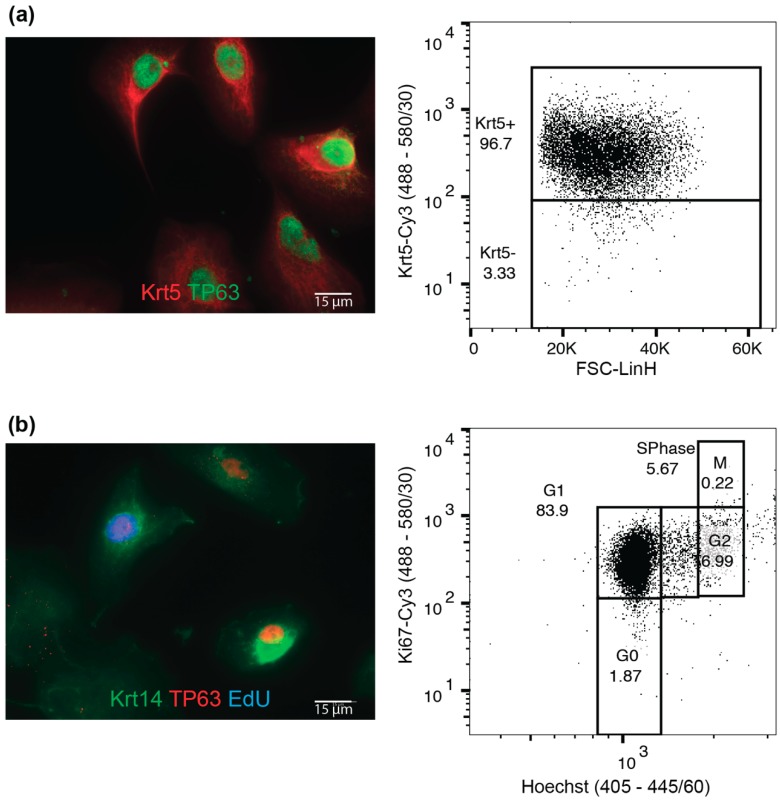
Maintenance of macaque basal cell (BC) cultures. (**a**) Macaque BC plated on chamber slides and stained for basal cell marker Krt14 and TP63, and flow cytometric analysis for Krt5expression; and (**b**) macaque BCs treated for 2h with EdU and stained for basal cell markers Krt14 and TP63, and Click-It EdU to visualize proliferative cells; flow cytometric analysis for DNA content and Ki67 to asses cell cycle properties of cells. Scale bars are 15 µm.

**Figure 3 mps-02-00079-f003:**
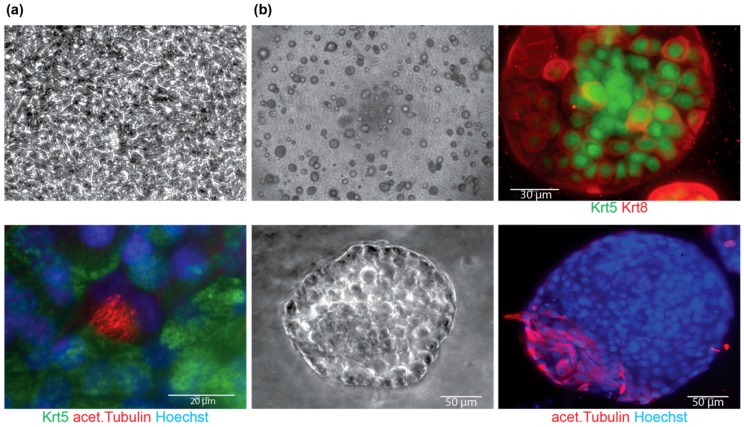
Differentiation of macaque BC cultures. (**a**) Macaque BC culture on cell culture inserts before transition to ALI with cobble stone morphology (top) and 14 days after transition into ALI, showing differentiated cells by acet. Tubulin staining (bottom), scale bar is 20 µm; and (**b**) bright-field images of tracheosphere cultures at 4× (left panel top) and 20× (left panel, bottom, scale bar is 50 µm) and stained for markers of differentiation Krt8 (top, right panel, scale bar is 30 µm) and acet. Tubulin (bottom, right panel, scale bar is 50 µm).

**Figure 4 mps-02-00079-f004:**
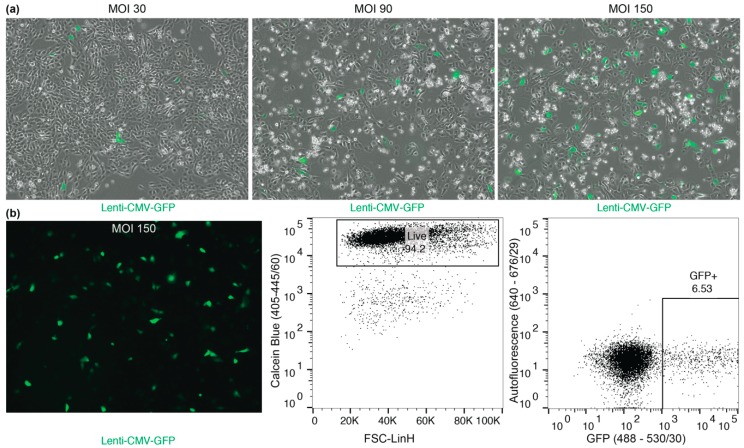
Transduction of macaque BCs using Lenti-CMV-GFP [10]. (**a**) Bright-field image, overlaid with green fluorescence; low to high MOI transduction (from left to right, MOI 30, MOI 90 and MOI 150 Lenti-CMV-GFP) of adherent macaque BCs, 72 h after infection. (**b**) Transduced cultures can be sorted with high viability (94.2%) for GFP expression.

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
