# Peer review of "Isolation, Maintenance and Differentiation of Primary Tracheal Basal Cells from Adult Rhesus Macaque"

_mps, 2019, doi:10.3390/mps2040079_

Round 1

Reviewer 1 Report

In this methods manuscript, Engler et al. describe a culture system for primarily isolated rhesus macaque lung basal cells. They go on to show that they can passage, freeze, and sort these cells while maintaining high viability up to 7 passages. Finally, they demonstrate that these basal cells can be infected with lentiviruses and will differentiate into appropriate airway cell types at air-liquid interface and in tracheospheres. While all of these methods have been described in other systems, including mouse and human, the manuscript is well written and includes a range of ages of animals with extensive information about expected yields and troubleshooting. The figures are well-organized and the protocol is well written. I feel that this protocol is essentially ready for publication as currently constructed. My only constructive comment is that Crispr-Cas9 manipulation is mentioned in section 4.3 and no data is shown or referenced, and I would either add some of these data or amend the section to avoid mentioning Crispr as the lack of data to support this point is confusing.

Author Response

We thank Reviewer 1 for their careful reading and constructive comments.

"In this methods manuscript, Engler et al. describe a culture system for primarily isolated rhesus macaque lung basal cells. They go on to show that they can passage, freeze, and sort these cells while maintaining high viability up to 7 passages. Finally, they demonstrate that these basal cells can be infected with lentiviruses and will differentiate into appropriate airway cell types at air-liquid interface and in tracheospheres. While all of these methods have been described in other systems, including mouse and human, the manuscript is well written and includes a range of ages of animals with extensive information about expected yields and troubleshooting. The figures are well-organized and the protocol is well written. I feel that this protocol is essentially ready for publication as currently constructed."

My only constructive comment is that Crispr-Cas9 manipulation is mentioned in section 4.3 and no data is shown or referenced, and I would either add some of these data or amend the section to avoid mentioning Crispr as the lack of data to support this point is confusing.

 We have removed the text about Crispr-Cas9 manipulation from the manuscript to avoid any confusion. The amended paragraph is highlighted in yellow.

Reviewer 2 Report

Engler et al., provide a detailed description of isolating and maintaining primary tracheal basal cells from adult rhesus macques. Such protocols exist for mouse and human samples. Thus, I question the need for this methods paper on Rhesus Macques. 

Rationale for all the sub-sections/steps should be explained better and rationalized to the readership. An introductory sentence/paragraph would help. 

Lenti-virus section (3.4.3) please provide better rationale and more detail.

Long term storage is not addressed.

Minor:

Reagent set up should come first?

Grammatically, lack of full stops throughout. 

Include scale bars and better resolution image (fig 1). 

Author Response

We thank Reviewer 2 for careful reading and their comments.

Engler et al., provide a detailed description of isolating and maintaining primary tracheal basal cells from adult rhesus macaques. Such protocols exist for mouse and human samples. Thus, I question the need for this methods paper on Rhesus Macaques. 

We understand that protocols exist for mouse and human basal cells. However, the conditions for growing either mouse or human cells do not support the growth of mmuBCs. We assert that this protocol for the maintenance and expansion of macaque basal cells will facilitate the development of preclinical, large animal models of cell replacement therapy.

Rationale for all the sub-sections/steps should be explained better and rationalized to the readership. An introductory sentence/paragraph would help. 
We have added a text section explaining the rational in the introductory paragraph under 3.0 before 3.1. Changes in response to comments from Reviewer 2 are highlighted in green

Lenti-virus section (3.4.3) please provide better rationale and more detail.
We have amended the text to reflect the Reviewers critique and highlighted the text portion.

Long term storage is not addressed.
We thank the reviewer for identifying this oversight. We have added the missing information under 3.5.1. and highlighted the new text.

Minor points:
Reagent set up should come first?
In regards to the positioning of the reagents setup we are in line with the template provided by MDPI for the MP formatting.

Grammatically, lack of full stops throughout. 
We have read through the manuscript carefully and applied full stops.

Include scale bars and better resolution image (fig 1). 
We have amended Figure 1 in accordance with the Reviewers suggestions.